

# A variable-rate quantitative trait evolution model using penalized-likelihood

Liam J. Revell[1,2]

[1] Department of Biology, University of Massachusetts Boston, Boston, MA, USA
[2] Facultad de Ciencias, Universidad Católica de la Santísima Concepción, Concepción, Chile

## ABSTRACT

In recent years it has become increasingly popular to use phylogenetic comparative methods to investigate heterogeneity in the rate or process of quantitative trait evolution across the branches or clades of a phylogenetic tree. Here, I present a new method for modeling variability in the rate of evolution of a continuously-valued character trait on a reconstructed phylogeny. The underlying model of evolution is stochastic diffusion (Brownian motion), but in which the instantaneous diffusion rate ($\sigma^2$) *also* evolves by Brownian motion on a logarithmic scale. Unfortunately, it's not possible to simultaneously estimate the rates of evolution along each edge of the tree *and* the rate of evolution of $\sigma^2$ itself using Maximum Likelihood. As such, I propose a penalized-likelihood method in which the penalty term is equal to the log-transformed probability density of the rates under a Brownian model, multiplied by a 'smoothing' coefficient, $\lambda$, selected by the user. $\lambda$ determines the magnitude of penalty that's applied to rate variation between edges. Lower values of $\lambda$ penalize rate variation relatively little; whereas larger $\lambda$ values result in minimal rate variation among edges of the tree in the fitted model, eventually converging on a single value of $\sigma^2$ for all of the branches of the tree. In addition to presenting this model here, I have also implemented it as part of my *phytools* R package in the function *multirateBM*. Using different values of the penalty coefficient, $\lambda$, I fit the model to simulated data with: Brownian rate variation among edges (the model assumption); uncorrelated rate variation; rate changes that occur in discrete places on the tree; and no rate variation at all among the branches of the phylogeny. I then compare the estimated values of $\sigma^2$ to their known true values. In addition, I use the method to analyze a simple empirical dataset of body mass evolution in mammals. Finally, I discuss the relationship between the method of this article and other models from the phylogenetic comparative methods and finance literature, as well as some applications and limitations of the approach.

# INTRODUCTION

As the quantity and quality of data for phylogenetic inference multiplies rapidly in the current genomic age (*Philippe et al., 2005*), so too has grown the popularity of using estimated phylogenetic trees to make inferences about the history of life on Earth (*Harvey, 1996*; *Nunn, 2011*; *O'Meara, 2012*). This endeavor, usually referred to as phylogenetic comparative biology, involves taking an evolutionary tree or set of trees obtained from

Corresponding author
Liam J. Revell, liam.revell@umb.edu

phylogenetic inference, often combined with phenotypic trait data for the species in the tree, and then making some kind of quantitative inference about evolution (*Garamszegi, 2014*; *Harmon, 2019*). A wide variety of different methodologies now comprise the field of phylogenetic comparative biology. These are often collectively referred to as 'phylogenetic comparative methods' (*Felsenstein, 1985*; *Nunn, 2011*; *O'Meara, 2012*; *Garamszegi, 2014*; *Harmon, 2019*).

A significant advance that's occurred over about the past 15 years has been the development and implementation in software of statistical methodologies that allow the user to model, or explicitly take into consideration, heterogeneity in the tempo and mode of evolution across the branches or clades of our evolutionary tree (*e.g.*, *Butler & King, 2004*; summarized in *Harmon, 2019*). For instance, in a now classic paper, *O'Meara et al. (2006*; also see *Thomas, Freckleton & Székely, 2006*) published an innovative model in which the rate of evolutionary change for a trait (normally denominated $\sigma^2$ in this type of study) was permitted to differ between different parts of the phylogeny, such as specific branches or clades, as determined *a priori* by the investigator. In the years since the publication of that seminal paper, a wide variety of different related approaches have been developed and implemented in software, such as methods that permit the process or mode of evolution, and not just its tempo, to differ in different parts of a reconstructed phylogeny (*e.g.*, *Revell & Harmon, 2008*; *Revell & Collar, 2009*; *Revell et al., 2012*; *Beaulieu et al., 2012*; *Uyeda & Harmon, 2014*; *Caetano & Harmon, 2017*).

The predominant model that's been used to study quantitative trait evolution on phylogenetic trees is called *Brownian motion* (*Felsenstein, 1985*; *O'Meara et al., 2006*; *Revell, Harmon & Collar, 2008*; *Harmon, 2019*). Brownian motion is a model of stochastic diffusion in which the trait changes randomly and continuously through time. The expected value of the trait is constant and equal to the starting value, $x_0$; and the *variance* (either between two or more lineages diverging by the process, or between the current state of a lineage and its ancestor) increases linearly with a rate of $\sigma^2$ (*O'Meara et al., 2006*; *Harmon, 2019*). Because $\sigma^2$ is the rate of increase in variance with time under our evolutionary process, our *estimate* of $\sigma^2$ is typically referred to as the *rate of evolution* of the trait (*O'Meara et al., 2006*). Brownian motion is described in much greater detail in other articles and books, such as *Felsenstein (2004)*, *O'Meara et al. (2006)*, *Revell, Harmon & Collar (2008)*, and *Harmon (2019)*.

Herein, I propose a new method for modeling variation in the *rate* of character evolution across the branches and clades of a phylogeny. Under this model, we assume that Brownian evolution has a different value ($\sigma_i^2$) on each of the branches of the tree, and that the log-values of these rates, in turn, have evolved *via* a separate Brownian process, also called *geometric Brownian motion*, theoretically with a separate rate, $\sigma_{BM}^2$—although I'll describe later why it may not be possible to estimate this rate. I will show how to fit this model to data using a penalized-likelihood approach (*Sanderson, 2002*) in which the penalty term is proportional to the logarithm of the probability of the log rates ($\log(\sigma_i^2)$) across edges under our model. I'll also apply the model to a classic empirical dataset for mammalian body size evolution from *Garland, Harvey & Ives (1992)*, and discuss both applications and limitations of the approach. I envision the primary use of this method to

be in *exploratory* data analysis—although it may also be possible to employ it in examining specific alternative hypotheses for evolutionary change through time.

## METHODS AND RESULTS

In this section I'll begin by introducing our model of evolution, then I'll apply it to several different simulated datasets, and, finally, I'll use the method to analyze a simple empirical dataset and tree.

### The model

The model of this article is as follows. We assume that the phenotypic trait of interest, $x$, evolves by a process of Brownian motion (*Felsenstein, 1985*; *O'Meara et al., 2006*; *Harmon, 2019*). Under standard Brownian motion evolution, changes are random and normally distributed, with a mean value of 0 and a variance of $\sigma^2 t$, in which $\sigma^2$ is the instantaneous variance of the Brownian process and $t$ is the elapsed time over which change is presumed to have occurred (*Felsenstein, 2004*; *Harmon, 2019*).

In this scenario, we expect that the values for our trait at the tips of the phylogeny will have a multivariate normal distribution with an expectation equal to the value of the trait at the start of the process, $x_0$, and a variance-covariance matrix given by $\sigma^2 \mathbf{C}$ (*e.g.*, *O'Meara et al., 2006*; *Revell & Collar, 2009*). Here, $\mathbf{C}$ is an $n \times n$ matrix for $n$ species containing the height above the root of the common ancestor of each pair of terminal taxa, $i$ and $j$, in the matrix position $C_{i,j}$ (*e.g.*, *Rohlf, 2001*; *Revell, Harmon & Collar, 2008*). $\sigma^2$ is the rate of evolution, as previously defined.

With just data and our phylogeny, we can find Maximum Likelihood estimates of the two parameters of the Brownian model, $\sigma^2$ and $x_0$, simply by maximizing the following expression giving the likelihood (*O'Meara et al., 2006*). This equation may be familiar to many readers because it's based on the multivariate normal probability density (*O'Meara et al., 2006*; *Revell & Harmon, 2008*).

$$l(\sigma^2, x_0|\mathbf{x}, \mathbf{C}) = \frac{\exp\left[-\frac{1}{2}(\mathbf{x} - \mathbf{1}x_0)'(\sigma^2\mathbf{C})^{-1}(\mathbf{x} - \mathbf{1}x_0)\right]}{\sqrt{2\pi^n|\sigma^2\mathbf{C}|}} \tag{1}$$

In our Eq. (1), $\mathbf{x}$ is an $n \times 1$ vector containing species values of $x$ for each of the $n$ taxa in our phylogeny. $\mathbf{1}$ is an $n \times 1$ vector of 1.0 s. Finally, $|\sigma^2\mathbf{C}|$ is the determinant of $\sigma^2\mathbf{C}$.

On a log-scale, this is equivalent to the following—in which $L$ (now and henceforward) is used to indicate the *log*-likelihood, as opposed to the likelihood on its original scale.

$$L = -(\mathbf{x} - \mathbf{1}x_0)'(\sigma^2\mathbf{C})^{-1}(\mathbf{x} - \mathbf{1}x_0)/2 - \log(|\sigma^2\mathbf{C}|)/2 - \log(2\pi^n)/2 \tag{2}$$

According to my penalized-likelihood approach, instead of computing the log probability density of just the species data at the tips of the tree under our model, we allow each edge of the tree to have a different value of the Brownian rate parameter, $\sigma^2$. Then we compute the penalized log-likelihood of the set of rates ($\sigma_0^2$, $\sigma_1^2$, $\sigma_2^2$, and so on), and the root state of our trait ($x_0$), given our tip data and reconstructed phylogeny.

We do this by maximizing the following expression—in which the penalty coefficient, $\lambda$, is set by the user, rather than estimated from the data. This is exactly equal to the log-likelihood of our model of evolution given the data; *minus* the negative log probability density of the log rates ($\log(\sigma_0^2)$, $\log(\sigma_1^2)$. $\log(\sigma_2^2)$, etc.) under a model in which the logarithm of the rate of evolution evolves by Brownian motion.

$$
\begin{aligned}
L(\sigma_0^2, \sigma_1^2, \ldots, x_0 | \mathbf{x}, \mathbf{C}_{ext}, \lambda) = &-(\mathbf{x} - \mathbf{1}x_0)'\mathbf{T}^{-1}(\mathbf{x} - \mathbf{1}x_0)/2 - \log(|\mathbf{T}|)/2 - \log(2\pi^n)/2 \\
&-\lambda(\mathbf{s} - \mathbf{1}s_0)'\mathbf{C}_{ext}^{-1}(\mathbf{s} - \mathbf{1}s_0)/2 - \lambda \log(|\mathbf{C}_{ext}|)/2 - \lambda \log(2\pi^{n+m-1})/2
\end{aligned}
\tag{3}
$$

Reviewing Eq. (3), we can see that the first part of the expression is almost exactly the same as Eq. (2), *except* that we've substituted $\mathbf{T}$ for $\sigma^2\mathbf{C}$. The second part is *also* similar (because it's based on a multivariate normal probability density), and gives $\lambda$ times the negative log probability density of the set of values for $\sigma_i^2$ at all of the nodes and tips of our phylogenetic tree. This latter part of the equation is our 'roughness penalty' (*e.g.*, *Sanderson, 2002*) as it *increases* for a given value of $\lambda$ (that is, penalizing the likelihood more) when $\sigma^2$ varies more from edge to edge, or decreases when it varies less.

More specifically, in Eq. (3), $\mathbf{T}$ is a matrix giving the sum of the edge lengths leading from the root to each common ancestor of each pair of species, $i$ and $j$ (in position $T_{i,j}$), multiplied by the branch-specific rates for each corresponding edge on that path (Fig. 1). $\mathbf{C}_{ext}$ is an *extended* version of the matrix $\mathbf{C}$, above, but of dimension $n + m - 1$ for $n$ species and $m$ internal nodes. $\mathbf{s}$ is an $n + m - 1 \times 1$ vector of the logarithms of each rate at each node or terminal taxon, except the root node (that is to say, $\mathbf{s} = [\log(\sigma_1^2), \log(\sigma_2^2), \ldots, \log(\sigma_{n+m-1}^2)]$). Lastly, $s_0$ is the logarithm of the rate of evolution at the root of the tree, $\log(\sigma_0^2)$. Other terms are defined as in Eqs. (1) and (2).

Although we compute $\mathbf{T}$ by multiplying the *average* rate of evolution for each edge in the tree by the length of the edge, as noted above, and then by summing these quantities up the phylogeny, our penalized-likelihood Eq. (3) is given in terms of *node* and *tip* (as opposed to edge) values of $\log(\sigma_i^2)$. As such, to calculate $\mathbf{T}$ we need to first compute the average rate for each edge as a function of the rates at the nodes that subtend it. To do this, we integrate the log-linear function between each pair of ancestor-descendant nodes (or between an internal node and a tip, for a terminal edge of the phylogeny). This is done by calculating $a(b)/b - a/b$, in which $a = \sigma_2^2$ and $b = \log(\sigma_1^2) - \log(\sigma_2^2)$ for each pair of two rates ($\sigma_1^2$ and $\sigma_2^2$, in this example) that begin and end a specific edge of our tree.

We define $\mathbf{C}_{ext}$ as a matrix of dimension $n + m - 1 \times n + m - 1$ containing the height above the root of the most recent common ancestor of every pair of terminal taxa (of which the tree contains $n$) and internal nodes (of which the tree contains $m$), except for the global root of the tree (hence the $- 1$). This is proportional to the expected variance-covariance matrix of the tips and internal nodes under a Brownian evolutionary process (*O'Meara et al., 2006*; *Harmon, 2019*).

Inspection of Eq. (3) leads us to some predictions about how our estimation method should behave as a function of the smoothing parameter value, $\lambda$. Firstly, as $\lambda$ is increased by the user we expect that the estimated rate of evolution, $\sigma^2$, should converge towards a single value for all edges of the phylogeny—and, furthermore, that this value of $\sigma^2$ will

a)

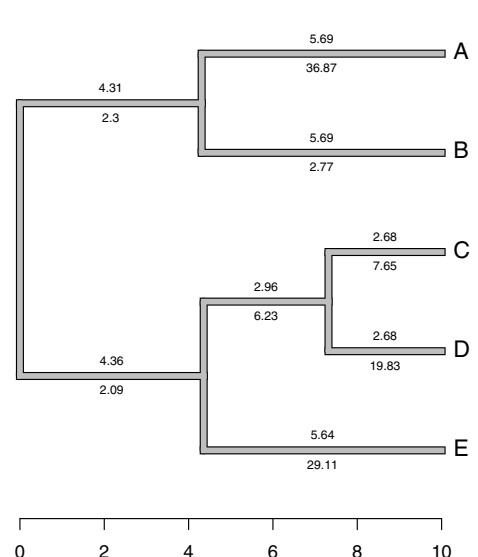

b)

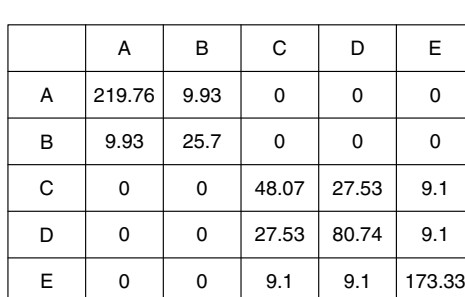

| | A | B | C | D | E |
|---|---|---|---|---|---|
| A | 219.76 | 9.93 | 0 | 0 | 0 |
| B | 9.93 | 25.7 | 0 | 0 | 0 |
| C | 0 | 0 | 48.07 | 27.53 | 9.1 |
| D | 0 | 0 | 27.53 | 80.74 | 9.1 |
| E | 0 | 0 | 9.1 | 9.1 | 173.33 |

**Figure 1** (A) Simple phylogeny of five taxa with the total edge length (above each branch) and $\sigma^2$ rates (below it); (B) calculation of the expected variance covariance matrix of tip values for the trait $x$, T, given the branch length and rates of panel (A). Each $i$, $j$th element corresponds to the sum of the products above and below each edge along all the branches leading from the root to the common ancestor of taxa $i$ and $j$.

also be the Maximum Likelihood Estimate (MLE) of $\sigma^2$ in a single-rate Brownian evolution model. Secondly, we conversely expect that as $\lambda$ is decreased towards 0, $\sigma^2$ should vary freely from edge to edge in the tree. If the rate of evolution does indeed evolve by geometric Brownian motion, then some intermediate value of $\lambda$ will lead to our best estimates of the values of $\sigma^2$ for each edge; however, this specific 'best' value of $\lambda$ is very likely to differ from case to case. I'll discuss possible strategies for selecting appropriate values of $\lambda$ in a subsequent section.

## Simulated example 1: Brownian rates

For the first example of applying the method of this article, I will suppose that evolution of our phenotypic trait, $x$, occurs by Brownian motion, and that the logarithm of the rate of evolution *also* evolves in a Brownian fashion (*i.e.*, by geometric Brownian motion, see above). This type of evolution can be seen in Fig. 2.

For panel (A) of the figure, I simulated different rates of Brownian evolution for each edge of the tree under our model. In panel (B), I've obtained a trait vector (**x**) based on the rates simulated for panel (A). Then, I projected my phylogeny into this trait space using a visualization technique called a 'traitgram' plot (*Schluter et al., 1997*; *Evans et al., 2009*; *Revell, 2013*), but in which each edge has been colored by the generating rate. From the figure it can be pretty clearly seen that *redder* branches, with higher evolutionary rates in panel (A), tend to be associated with *larger* changes on the traitgram of panel (B)—precisely as one would expect under our model (Fig. 2).
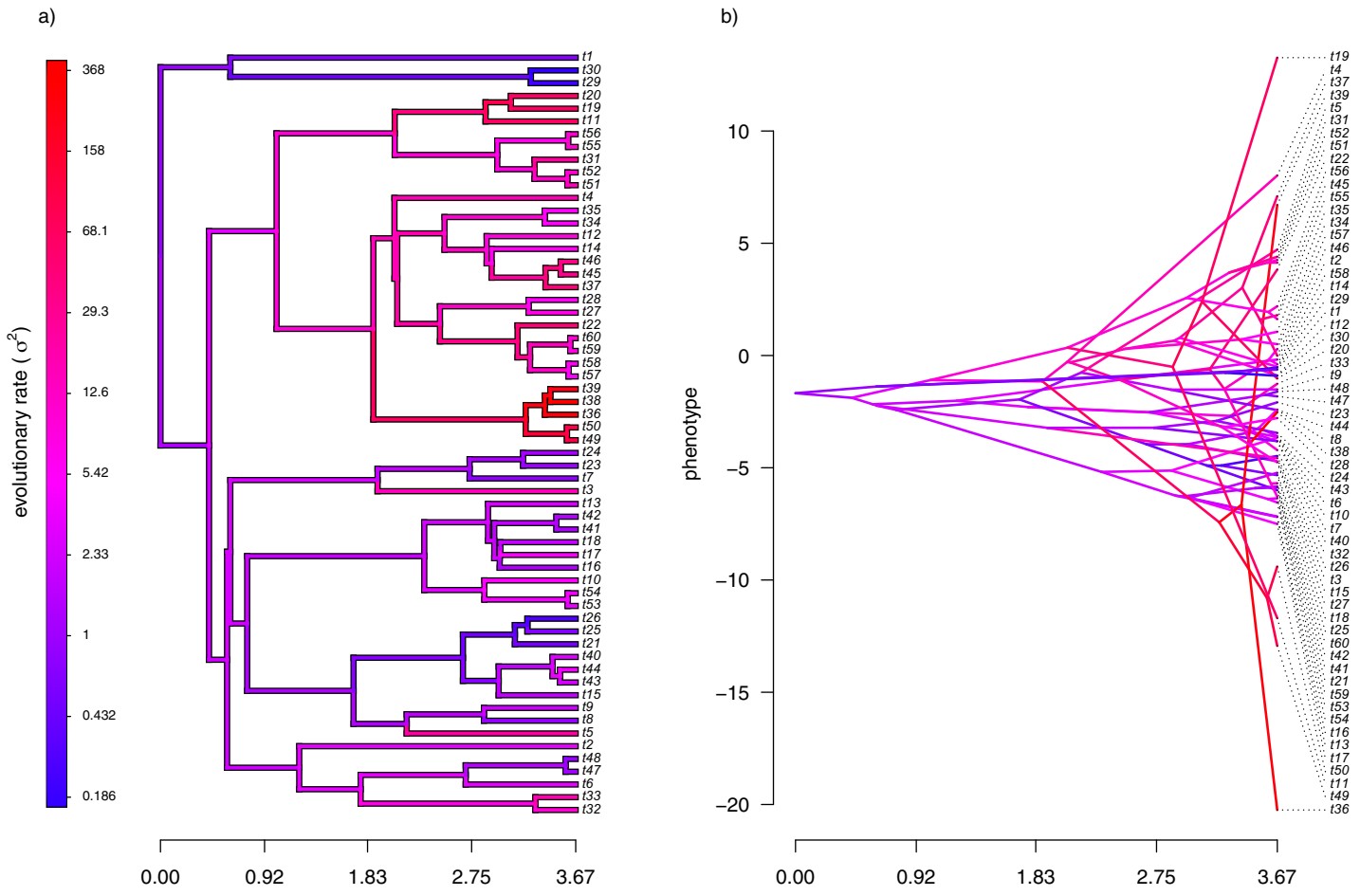

**Figure 2** (A) Simulated evolutionary rates, σ², in which the logarithm of the rate of evolution evolves by Brownian motion on the tree; (B) the phylogeny of panel (A) projected into a space defined by time (on the horizontal axis) and a simulated trait vector, x, obtained using the rates of panel (A).

In my simulation, rate variation among edges varies by more than three orders of magnitude: from $\sigma^2 < 0.2$ to $\sigma^2 = 368$. Although I generated these data mainly for illustrative purposes, this magnitude of rate variation among edges or clades is not entirely inconsistent with what has been found in empirical studies. For instance, *Puttick, Thomas & Benton (2014)* measured a rate of body size evolution more than 9,000 times higher than the background rate on the edge leading to extant avian reptiles. Using different methodology, *Rabosky et al. (2014)* found a nearly twenty-fold difference in the rate of body shape evolution between different lizard clades. Finally, *Martin (2016)* found more than a thousand-fold difference in the rate of jaw morphology evolution in Caribbean pupfishes (genus *Cyprinidon*). On the other hand, not every study looking for rate variation finds it. *LeRoy et al. (2019)*, for example, found relatively little evidence for heterogeneity in the evolutionary rate of leaf litter decomposition rates across a phylogeny of 239 species of angiosperms.

With the simulated phylogeny and data in hand, we can then proceed to fit a rate-variable model to the data of Fig. 2 using our approach of penalized-likelihood.

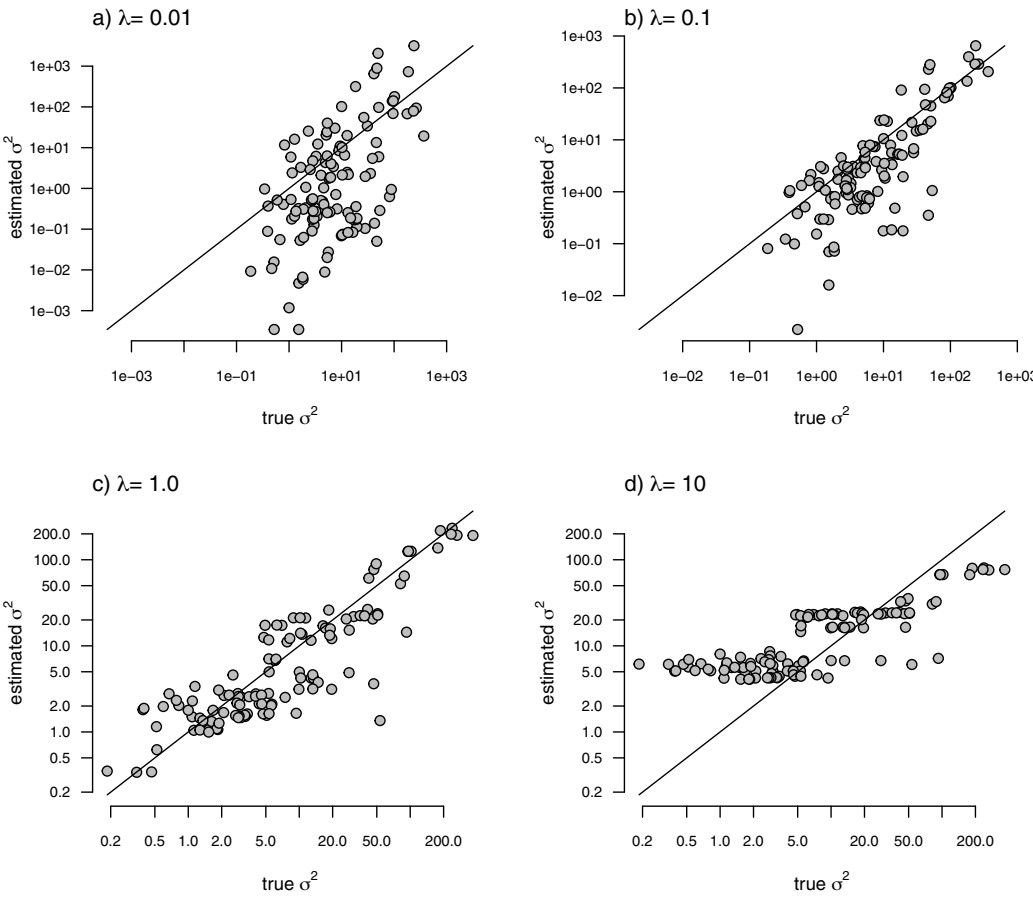

**Figure 3 Estimated values of $\sigma^2$ ($y$) compared to their known true values ($x$) for the data of Fig. 2 obtained using the penalized-likelihood approach with four different values of the penalty term coefficient, $\lambda$: (A) $\lambda = 0.01$; (B) $\lambda = 0.1$; (C) $\lambda = 1$; and (D) $\lambda = 10$.** The 1:1 line is indicated.

Our penalized-likelihood method, remember, requires that we select a value of $\lambda$, the penalty coefficient or 'smoothing' parameter. Higher values of $\lambda$ correspond to more smoothing, such that adjacent branches of the tree will tend to have more similar values of $\sigma^2$. If we do this for four different values of $\lambda$ ($\lambda$ = 0.01, 0.1, 1, and 10), and then each time compare our estimated rates to their known true values from Fig. 2, we obtain the result shown in Fig. 3.

We can see that the estimated values of $\sigma^2$ are in all cases quite strongly correlated with their generating values; however, (at least here) the strongest relationship is for an 'intermediate' value of $\lambda$ ($\lambda$ = 1), on the range that we chose to use. Readers should keep in mind, of course, that this should not generally be expected to be the case even if our rates have indeed evolved under the assumed process. I'll explain more about selecting appropriate values of $\lambda$ in the Discussion.

### Simulated example 2: uncorrelated rates

In addition to generating data under our assumed model of Brownian evolution of the rates on a log scale, I also simulated *uncorrelated* rate variation among edges. In this case,
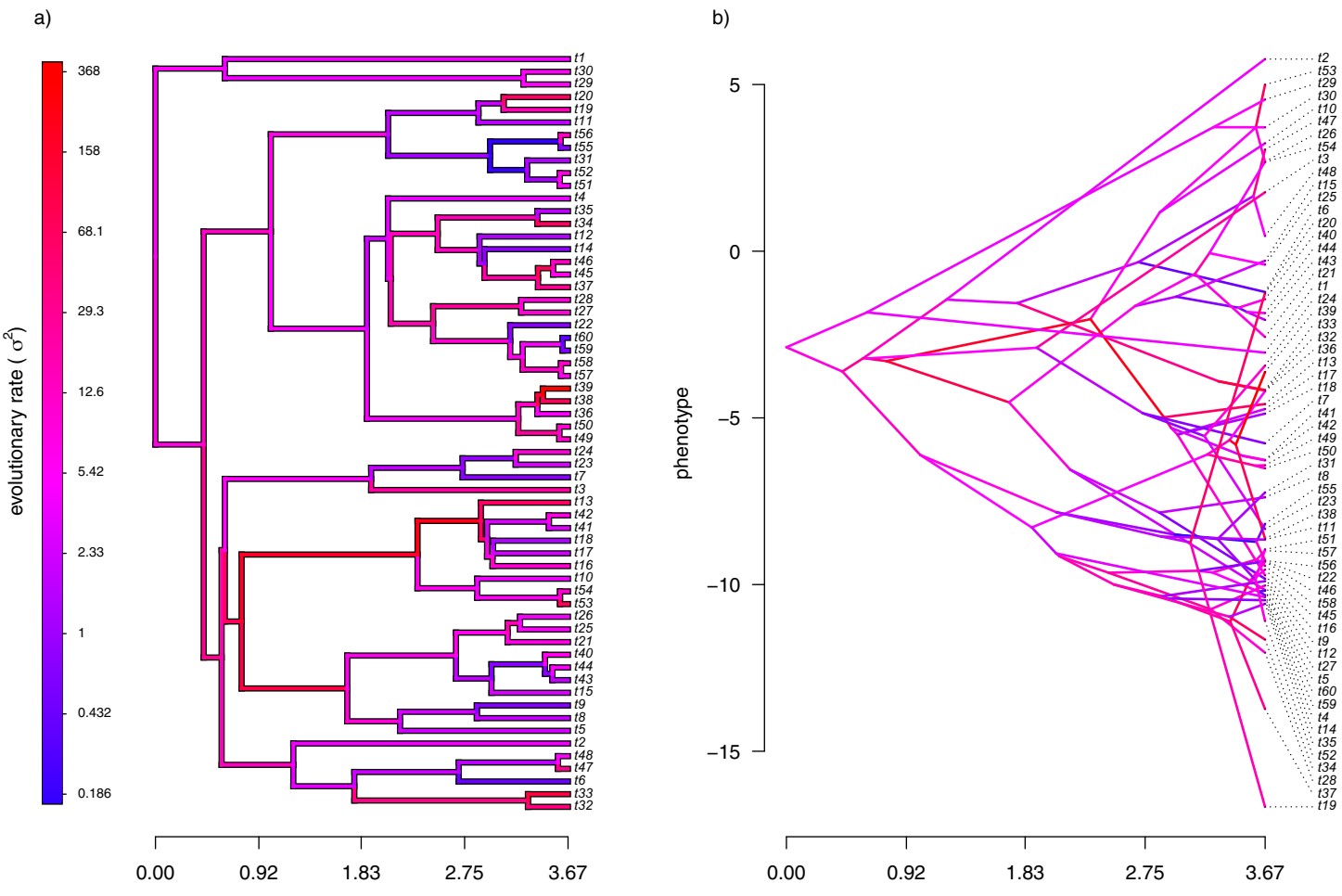

**Figure 4** (A) Simulated evolutionary rates, $\sigma^2$, in which the logarithm of the rate of evolution is uncorrelated between nodes and tips on the phylogeny; (B) the tree projected into a space defined by time (on the horizontal axis) and a simulated trait vector, x, obtained using the rates of panel (A).

I simply took the values simulated in example 1, above, and randomized them among nodes. This resulted in exactly the same range of simulated range of values for $\sigma^2$ as in Fig. 2, but with rates that were uncorrelated among different parts of the tree. To simulate data on the tree using my new rates, I simply computed the mean rate for the subtending edge of each pair of nodes or tips using the same procedure as outlined above.

My simulated data for this example are given in Fig. 4. Just as in Fig. 2, above, panel (A) shows the simulated rates on each edge of the phylogeny, while panel (B) gives a projection of the tree into phenotype space in which the data vector used for the projection, **x**, was obtained using the rates of panel (A). It should be evident from both figure panels that in this case there is little autocorrelation of nearby rates of evolution on the tree, just as we'd expect based on our simulation procedure (Fig. 4).

With my phylogeny and these simulated data, I next repeated exactly the same analysis that I undertook for the data of Fig. 2 (once again, with four different values of $\lambda$: 0.01, 0.1, 1, and 10). I then, likewise, compared the *estimated* values of $\sigma^2$ to each of the values used in simulation. The results of this comparison are shown in Fig. 5. Strikingly, the
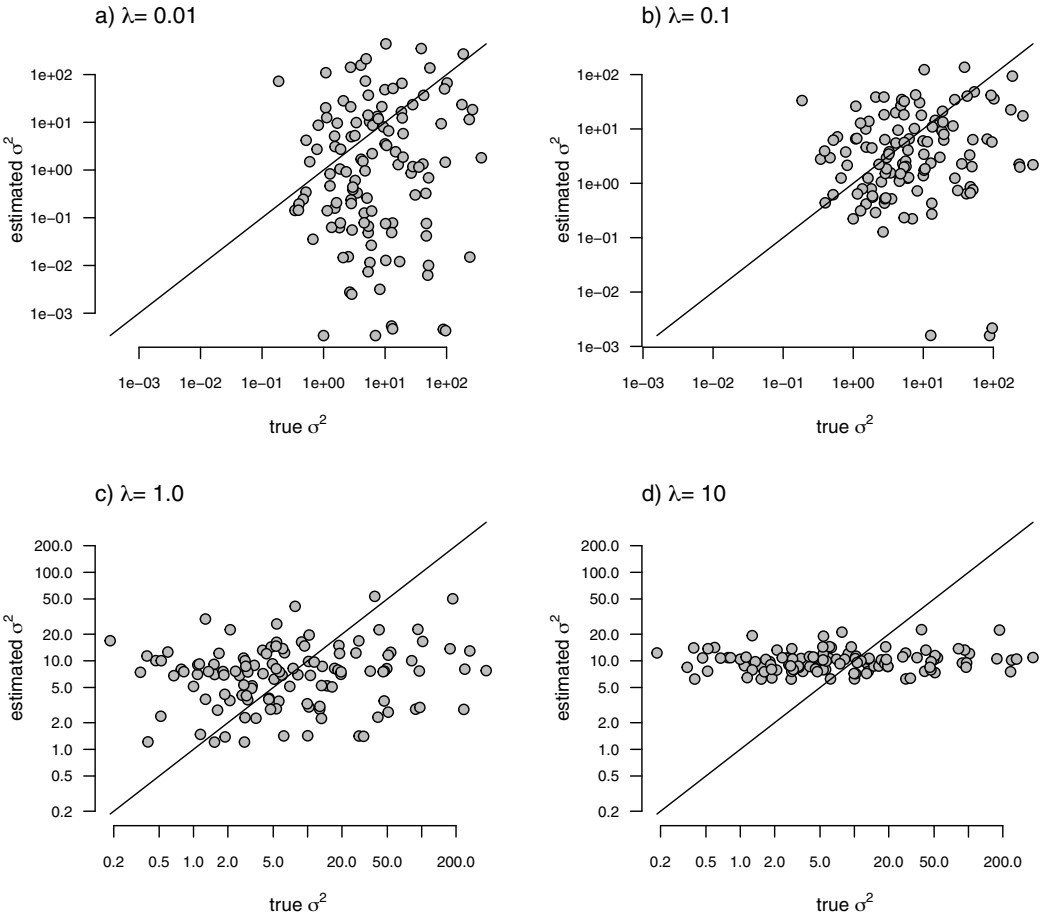

**Figure 5 Estimated values of $\sigma^2$ ($y$) compared to their known true values ($x$) for the data of Fig. 4 obtained using the penalized-likelihood approach using four different values of the penalty term coefficient, $\lambda$: (A) $\lambda = 0.01$; (B) $\lambda = 0.1$; (C) $\lambda = 1$; and (D) $\lambda = 10$.** The 1:1 line is indicated.

correlations between our generating and estimated values of $\sigma^2$ are much weaker than they were in Fig. 3 when the generating process for the simulated rates was the assumed model in our estimation method: geometric Brownian motion evolution.

## Simulated example 3: discrete rate shifts

In addition to Brownian evolution of $\sigma^2$ (from the first simulation example), and uncorrelated $\sigma^2$ (from the second), I also simulated discrete *shifts* in the rate of Brownian evolution on the phylogeny. This is analogous to the fitted model in *O'Meara et al. (2006*; also see *Thomas, Freckleton & Székely, 2006*; *Revell & Collar, 2009*; *Beaulieu et al., 2012)*. Here, however, when I fit my model using penalized-likelihood, I will not inform my model-fitting method of the specific locations of the rate shifts on our phylogeny.

To generate data of this type, I first simulated the evolution of discrete regimes on the same tree as I used to create the data of Figs. 2 and 4 under a continuous-time Markov chain (the M*k* model of (*Lewis, 2001*; *Harmon, 2019*)). Simply for the sake of convenience in subsequent calculations, I filtered any simulation in which more than one transition

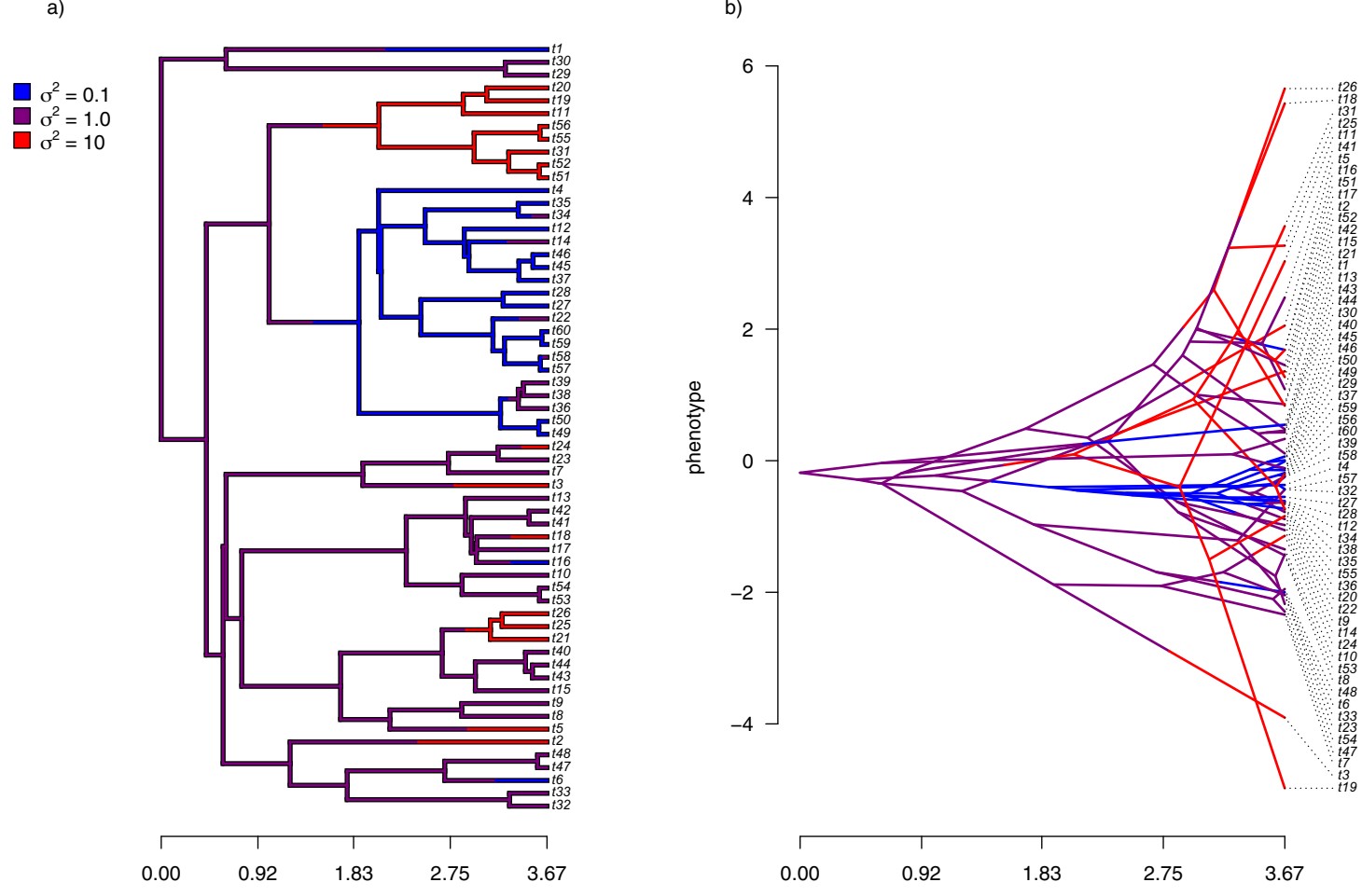

**Figure 6** (A) Simulated evolutionary rates, $\sigma^2$, in which rate of evolution shifts discretely under a continuous-time Markov process; (B) the tree of panel (A) projected into a space defined by time (on the horizontal axis) and a simulated trait vector, x, obtained using the rates of panel (A).               

between regimes occurred on a single branch. In addition, I moved the transition between regimes to the precise midpoint of each edge. Since our model for rate evolution is one in which nodes and tips have values of $\sigma^2$, this simply makes for an easier comparison between our simulation and our estimated rates. I also constrained the substitution model of my rate regimes to be *ordered*. That is, transitions were only allowed to occur from low rate to medium rate, and *vice versa*; and from medium to high rate, and *vice versa*—but not directly between the lowest and highest rate categories.

Having generated different stochastic regimes on the phylogeny, I then proceeded to simulate phenotypic trait evolution given these regimes. For my low, medium, and high rates, I arbitrarily set $\sigma^2$ to be equal to 0.1, 1, and 10, respectively. The resultant phylogeny and simulated data are given in Fig. 6.

Just as for the previous two examples, I fit our multi-rate Brownian motion model to these data using penalized-likelihood for four different values of the smoothing penalty coefficient, $\lambda$ ($\lambda$ = 0.01, 0.1, 1, and 10). I then compared the estimated values from our fitted models to the generating values from simulation. The results can be seen in Fig. 7.

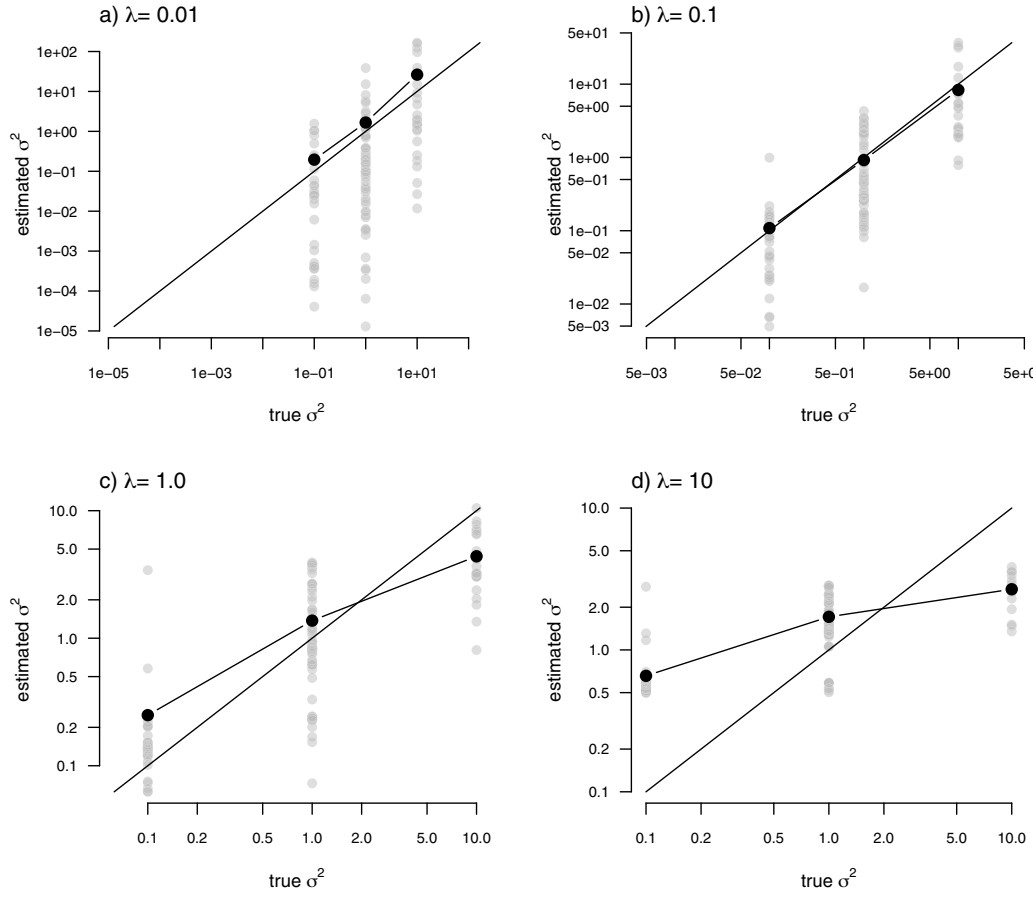

**Figure 7 Estimated values (grey points) of $\sigma^2$ ($y$) compared to their known true values ($x$) for the data of Fig. 6 obtained using the penalized-likelihood approach using four different values of the penalty term coefficient, $\lambda$: (A) $\lambda = 0.01$; (B) $\lambda = 0.1$; (C) $\lambda = 1$; and (D) $\lambda = 10$.** Black points give the mean estimated value of $\sigma^2$ for each simulated regime. The 1:1 line is also indicated.

One substantive difference between Fig. 7 and those prior to it is that in this case our nodes and tips all belong to one of only three, distinct rate categories: $\sigma^2 = 0.1$, 1, or 10. As such, I thought it would also be reasonable to compute the mean estimated value of $\sigma^2$ for each of the regimes, and then graph these mean values onto our plot. This analysis is also included in the figure. What's striking about the result is the close alignment between the mean estimated and generating values of $\sigma^2$, particularly for lower values of the smoothing parameter $\lambda$. On the other hand, many of the individual edge-specific $\sigma^2$ estimates are very *far* from the true, generating rate values (Fig. 7).

## Simulated example 4: invariant rates

For my last simulated example, I generated data for a single trait in which the rate of evolution, $\sigma^2$, was *constant* (and set to $\sigma^2 = 1$) over all the nodes and branches of the phylogeny. I then proceeded to fit a multi-rate model using penalized-likelihood, just as I did for all three of the previous simulations. Finally, I compared the estimated values, and

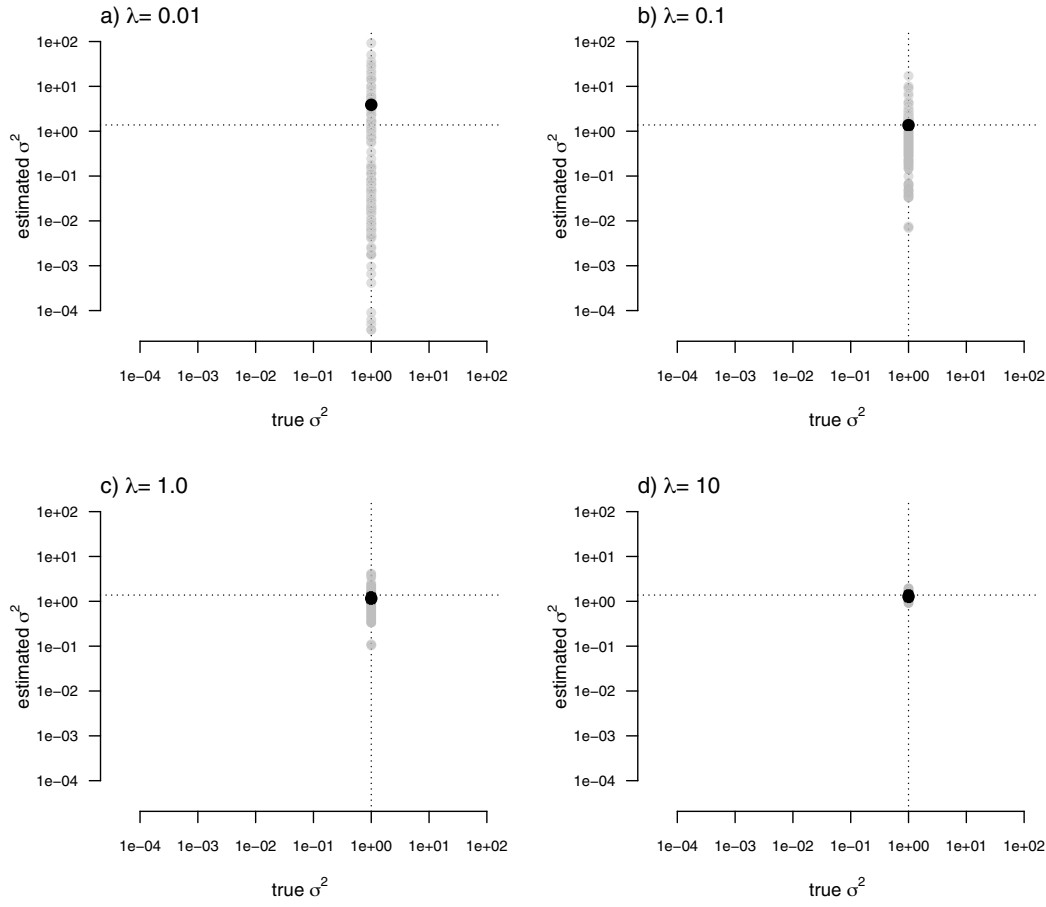

**Figure 8 Estimated values (grey points) of $\sigma^2$ ($y$) compared to their known true values ($x$) for data simulated with a constant rate of evolution, $\sigma^2 = 1$.** Vertical and horizontal lines show the true rate of evolution from the simulation, and the Maximum Likelihood Estimate of the rate obtained in a single-rate analysis (respectively). The black points in each panel shows the mean estimated value of $\sigma^2$ across all edges of the tree.

their means, to the known, *true* value of $\sigma^2$ for the same four levels of our smoothing coefficient, $\lambda$ ($\lambda$ = 0.01, 0.1, 1, and 10). The result of this analysis can be seen in Fig. 8.

In this figure, and in contrast to the equivalent plots from prior simulations, I've intentionally kept the $x$ and $y$ axes consistent between each figure panel. What the plots show is, firstly, that the *mean $\sigma^2$* across all edges of the phylogeny under our multi-rate penalized-likelihood analysis (shown using the heavy black dot on each panel) tends to be quite similar to the *true* single rate across each of the four values of $\lambda$ explored here, as well as to the MLE of $\sigma^2$ for a single rate model (shown using the horizontal dotted line; Fig. 8), as estimated using the *geiger* R package (*Pennell et al., 2014*). Secondly, the plot shows that for increasing values of $\lambda$, variation among edges of the tree decreases and eventually *converges* on the MLE for a single rate (Fig. 8). Both of these results are properties that we expected for our estimation procedure.

## Analysis of body mass evolution in mammals

Finally, I also analyzed a simple dataset for overall body mass evolution in a small phylogeny of mammals. These data, which consist of values for mass in kg transformed to the log scale and a phylogeny of 49 mammalian species, are from *Garland, Harvey & Ives (1992)* and come packaged with the R phylogenetics library *phytools* (*Revell, 2012*). I fit our variable-rate Brownian model with penalized-likelihood using the same four values of $\lambda$ as I employed with simulated data ($\lambda$ = 0.01, 0.1, 1, and 10). Then, I graphed the fitted models using the corresponding *phytools* plotting method for our fitted model object. This maps the estimated rates onto the edges of the tree and can use a color gradient that is specified by the user. Here, I used an inverted *heat.colors* gradient from the R package *grDevices* (*R Development Core Team, 2020*).

What's most notable about the result (shown in Fig. 9) is that estimated rates are highly correlated between analyses using different values of $\lambda$, even if their absolute magnitudes differ. Qualitatively, this analysis consistently reveals that the *highest* rates of body mass evolution in our tree are found among the Artiodactyla (even toed ungulates, such as deer, moose, bison, etc.), while the *lowest* rates of body mass evolution are observed for the Perissodactyla (odd-toed ungulates, such as horses and tapirs; Fig. 9).

## Notes on implementation

The model and methods of this study have been implemented for the R statistical computing environment (*R Development Core Team, 2020*), and all simulations and analyses were conducted in R. The penalized-likelihood method that I describe in the article is implemented as the function *multirateBM* of my *phytools* R package (*Revell, 2012*). *phytools* in turn depends on the important R phylogenetics packages *ape* (*Paradis & Schliep, 2019*) and *phangorn* (*Schliep, 2011*), as well as on a number of other R libraries (*Venables & Ripley, 2002*; *Ligges & Mächler, 2003*; *Lemon, 2006*; *Plummer et al., 2006*; *Chasalow, 2012*; *Becker et al., 2018*; *Gilbert & Varadhan, 2019*; *Azzalini & Genz, 2020*; *Qiu & Joe, 2020*; *Warnes, Bolker & Lumley, 2020*; *Goulet et al., 2021*; *Pinheiro et al., 2021*). Finally, some analyses in the latter part of the article were done with the help of the *geiger* package (*Pennell et al., 2014*).

## DISCUSSION

Phylogenetic comparative analysis is one of the leading weapons in the arsenal of evolutionary biologists interested in understanding and characterizing the history of life on this planet (*Nunn, 2011*; *Garamszegi, 2014*; *Harmon, 2019*). Many phylogenetic comparative methods have been developed in recent years, and a number of these are specifically designed to model heterogeneity in the tempo or mode of evolution among branches and clades of the tree (*e.g.*, *O'Meara et al., 2006*; *Revell & Harmon, 2008*; *Revell et al., 2012*; *Beaulieu et al., 2012*; *Uyeda & Harmon, 2014*; *Caetano & Harmon, 2017*).

In the present article, I describe a new phylogenetic comparative method—implemented as a function in my *phytools* R package (*Revell, 2012*). In this method, which I've developed to analyze the evolution of a single continuously-valued trait on the phylogeny, the rate of evolution ($\sigma^2$) is free to vary from branch to branch in the tree.

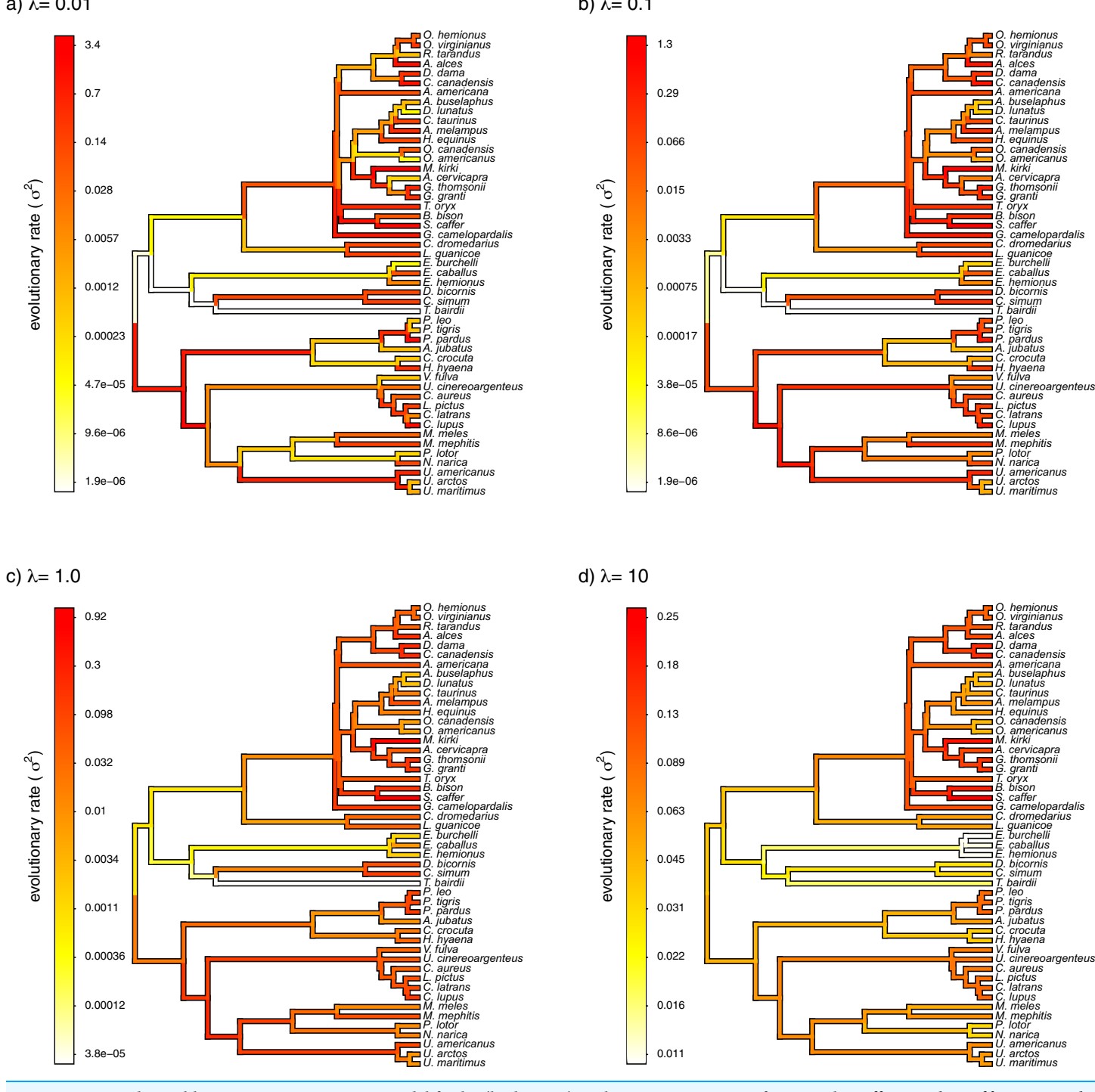

**Figure 9 Fitted variable-rate Brownian motion model for log(body mass) evolution in 49 species of mammals.** Different values of λ correspond to different penalty coefficients for rate variation among edges of the tree. Note that each panel has a different scale.

Model-fitting is done using the technique of penalized-likelihood, in which the penalty term is a function of the negative log probability density of the rates, assuming that the rates themselves evolve by a geometric Brownian process on the phylogeny.

I have applied this method to datasets simulated under a variety scenarios of potential interest to biologists that might consider using this approach with their own data. The first of these, in which the logarithm of the rate of evolution ($\log(\sigma^2)$) evolves by Brownian motion, is the underlying model that is *assumed* by the analysis method of this article. I show that for a range of different values of the smoothing parameter, $\lambda$, estimated evolutionary rates are correlated with the generating rates of the simulation (Fig. 3).

Under other simulation conditions we see different results. For instance, when the evolutionary rates are random and *uncorrelated* among edges, our estimated rates from the penalized-likelihood method also tend to be correlated with the generating rates used for simulation (Fig. 5)—though this correlation is much weaker than when the generating rates evolved *via* a geometric Brownian process. This behavior is understandable. When evolutionary rates are uncorrelated across the tree, there is simply not enough information in the tip data for a single character to reliably estimate the rates of evolution at each edge or node.

When data were generated with discrete rate shifts on the tree, the mean estimated rate across edges in each regime closely matched the generating value of the rate (Fig. 7). Finally, when there was *no* variation at all in rate among the edges of the phylogeny, the arithmetic mean estimated rate tended to quite closely match the MLE of $\sigma^2$ from a simple, one-rate model (Fig. 8).

In addition to this simulation, I applied the model to a classic empirical dataset of log body mass in 49 species of mammals (*Garland, Harvey & Ives, 1992*). Although the variation in estimated rates among edges decreased with increasing $\lambda$, the qualitative pattern of variation was quite similar among different $\lambda$ values—generally showing the highest rates of evolution of body mass in artiodactyls (even-toed ungulates, such as deer, cows, and antelopes), and the *lowest* estimated values of $\sigma^2$ for the perissodactyls (odd-toed ungulates, such as horses and tapirs).

## Why not just estimate $\sigma^2_{\mathrm{BM}}$ (or $\lambda$)?

Earlier in the article, I mentioned that it was probably not possible to simultaneously estimate the evolutionary rates for each edge or node in the tree ($\sigma^2_0$, $\sigma^2_1$, and so on) *and* the rate of evolution of the rates themselves (let's call it $\sigma^2_{BM}$); however, at the time I didn't say why this seemed likely to be the case. An imaginable (but ultimately untractable, as we'll see) solution to this problem would be to simply take the following expression for the likelihood (in which the first part is the probability density of our trait observations given our rates; and the second part is the probability of our rates, given an underlying model for their evolution) and maximize it.

$$l(\sigma^2_0, \sigma^2_1, \ldots, \sigma^2_{BM}, x_0 | \mathbf{x}, \mathbf{C}_{ext}) = \frac{\exp[-\frac{1}{2}(\mathbf{x} - \mathbf{1}x_0)'\mathbf{T}^{-1}(\mathbf{x} - \mathbf{1}x_0)]}{\sqrt{2\pi^n|\mathbf{T}|}}$$
$$\times \frac{\exp[-\frac{1}{2}(\mathbf{s} - \mathbf{1}s_0)'(\sigma^2_{BM}\mathbf{C}_{ext})^{-1}(\mathbf{s} - \mathbf{1}s_0)]}{\sqrt{2\pi^n|\sigma^2_{BM}\mathbf{C}_{ext}|}} \quad (4)$$

In Eq. (4), all the terms are as previously defined for Eqs. (1), (2), and (3); plus $\sigma^2_{BM}$ is the rate of geometric Brownian evolution of the rates ($s_0, s_1$, etc.), and **1** is a conformable vector of 1.0 s ($n \times 1$ in the first part of the equation, and $n + m - 1 \times 1$ in the second).

The problem with this logic, unfortunately, is that Eq. (4) will *always* go to $\infty$ as $\sigma^2_{BM} \rightarrow 0$. This is simply because a proper probability density function integrates to 1.0, and since the $s_i$s are not observed, the second part of Eq. (4) will have zero width and infinite height when all values of $s_i$ are equal and $\sigma^2_{BM} = 0$. A similar difficulty would apply to any scaling coefficient (such as $\lambda$) that multiplied the second term in our likelihood expression but not the first.

## How to choose a value of $\lambda$

I have so far presented a new, penalized-likelihood approach for modeling heterogeneity in the rate of evolution for a continuously-valued trait on the phylogeny. Unfortunately, since neither $\lambda$ nor the rate parameter of the Brownian evolutionary process of the rates are estimable, our method requires that we simply assign a value for the penalty or smoothing coefficient of our fitted model. Here, I employed the relatively simplistic tactic of trying multiple values of $\lambda$ for each simulated or empirical dataset, and then comparing the results. Analyses like those shown in Fig. 9 suggest that it might be possible to draw some credible inferences about evolution using this strategy.

Nonetheless, most readers are probably interested in a more rigorous approach to identifying reasonable values of the penalty or smoothing coefficient, $\lambda$. Although I've not yet explored it for the model of this paper, a tactic that seems to work well for other penalized-likelihood methods is *cross-validation* (*Stone, 1974*; *Efron & Gong, 1983*). In cross-validation we would first select a value of $\lambda$ and fit the model using this value. Next, we would successively drop one or more observations (tips) from our tree (either at random or one-by-one), re-estimate rates for all nodes, and then compare our estimated rates to their values from the model when it had been fit using the full dataset. Our preferred value for $\lambda$ would be the one that minimized the sum of squared differences between the measured rates in the global estimate and those computed from our subsampling procedure. This precise approach has proved successful in other rate-smoothing methods, including in phylogenetics (*Sanderson, 2002*; *Smith & O'Meara, 2012*). I've not yet implemented cross-validation in *phytools*, but it would not be difficult to script in R for *phytools* users already adept at such things. Unfortunately, model-fitting using the *phytools* function in which this method is implemented (*multirateBM*) is *extremely* time-consuming, and so this cross-validation procedure would take a very long time to run on even a single dataset as it requires that the model be fit to our data hundreds of times or more in one analysis (*Smith & O'Meara, 2012*).

## Relationship to other methods

The penalized-likelihood method of this article is related to various other phylogenetic methods designed to investigate how the rate or evolutionary process of a continuous character changes over time or in different parts of a phylogeny. A very important intellectual progenitor of the method is an approach developed by *O'Meara et al. (2006*;

also see *Thomas, Freckleton & Székely, 2006*), and implemented first in the software BROWNIE and subsequently in my *phytools* R package (*Revell, 2012*). According to the *O'Meara et al. (2006)* method, we first specify different regimes on the tree, and then we proceed to fit a Brownian motion model of evolution in which the rate of evolutionary change, $\sigma^2$, varies as a function of our mapped regimes. If each branch in the phylogeny was assigned to a unique regime, this would be equivalent to setting $\lambda = 0$ in our penalized-likelihood method—thus allowing $\sigma^2$ to change from branch to branch without penalty. (Although it's unlikely that the parameter estimates would be sensible in this case.)

A variety of other approaches also allow the rate of evolution to vary discretely in different parts of the phylogeny, but do not require that these regimes be specified *a priori* by the investigator. For instance, *Eastman et al. (2011)* developed a method to identify the locations of discrete Brownian evolution rate shifts on the phylogeny using Bayesian reversible-jump Markov chain Monte Carlo (rjMCMC). This method was initially implemented in the *auteur* R package, but then eventually incorporated into *geiger* (*Pennell et al., 2014*). Subsequently, *Uyeda & Harmon (2014)* developed a similar rjMCMC method, but extended it to evolutionary models other than Brownian motion, such as the well-known Ornstein-Uhlenbeck process (*Hansen, 1997*; *Butler & King, 2004*). This method is implemented in the R software *bayou* (*Uyeda, Eastman & Harmon, 2020*).

In parallel, *Revell et al. (2012)* developed a Bayesian method to identify discrete shifts in the evolutionary rate that used standard (rather than reversible-jump) MCMC. Likewise, *Venditti, Meade & Pagel (2011)* independently developed a similar rjMCMC method to that of *Eastman et al. (2011)* or *Uyeda & Harmon (2014)*, now implemented in the popular software *BayesTraits* (*Meade & Pagel, 2021*). Since when we average across the posterior sample of rates and rate regimes in any of these methods we can theoretically obtain a different rate for each edge of the phylogeny, their results will in some ways resemble those obtained from the penalized-likelihood approach I've presented herein. Nonetheless, these different MCMC and rjMCMC techniques invariably involve discrete (rather than continuous) changes in the evolutionary rate between (or along) branches of the phylogeny. Finally, the Bayesian MCMC phylogenetics software *BAMM*, by *Rabosky (2014)*, includes a method for detecting both discrete rate shifts and continuous changes in the evolutionary rate through time. This too is related to the approach that I have presented in the current article.

Separately from phylogenetic biology, the method of this paper is also closely related to an important model from the financial literature called a *SABR volatility model*, in which SABR is an acronym for 'stochastic α (alpha), β (beta), ρ (rho)' (*Hagan et al., 2002*). The SABR model is described by a set of stochastic differential equations in which the derivative or stock price changes through time under a diffusion process, but where the rate of diffusion *also* changes randomly with rate parameter α (*Hagan et al., 2002*). In fact, one could say that model of this article is a special case of SABR, but in which the β term of the model (which describes the skewness of stochastic volatility) and the ρ term (which describes the correlation between the stochastic term for change in price and α in the model) are both set to zero. Just as the model of this study reduces to a constant-rate Brownian evolution model as the penalty coefficient, $\lambda$, is increased, SABR too reduces to a

simpler model called the *constant elasticity of variance* (CEV) model (*Cox, 1975*; *Beckers, 1980*) if α (the volatility of stochastic volatity term) is fixed to 0. Seeing this connection between stochastic volatility models in finance, such as the SABR model, and the method of this study also opens up the possibility of adapting other parts of this class of model to phylogenetic comparative analysis, such as correlation between rate volatility ($\sigma^2_{BM}$ in our method) and the rate itself ($\sigma^2_i$). This would be equivalent to $\rho \neq 0$ in the SABR model.

One important difference between the model of this study and SABR, is that I have essentially assumed a single value of of $\sigma^2$ for each edge of the tree—which evolves from edge to edge under a process of geometric Brownian motion. I've then focused on estimating the rates, $\sigma^2_i$, for all the nodes and tips of the phylogeny, rather than the rate of change of the rate itself, $\sigma^2_{BM}$. This is because, as noted in a preceding section, allowing all of the $\sigma^2_i$ *and* $\sigma^2_{BM}$ to vary freely guarantees that when we maximize the expression for their joint likelihood (given by Eq. (4), above), all $\sigma^2_i$ will converge on a constant value and $\sigma^2_{BM}$ on zero. There are clues in the financial literature that it may be possible to calibrate our model and thus estimate $\sigma^2_{BM}$ (*e.g.*, *West, 2005*; *Fatone et al., 2013*; *Fatone et al., 2014*); however, I'll leave it to others cleverer than I to figure out *if* this is possible and how.

### What about fossils?

In both the simulations and empirical example I've focused on observations for only extant taxa. For better or for worse, this is extremely common in phylogenetic comparative biology (see (*Slater & Harmon, 2013*; *Blomberg, Rathnayake & Moreau, 2020*) for a discussion of the problem). In general, I would expect that the method of this paper will perform relatively well, even absent information from the fossil record, in identifying heterogeneity of the rate of evolution, $\sigma^2$, when this occurs between clades of relatively recently diverged lineages. On the other hand, under circumstances of both temporal and among-clade rate heterogeneity, the lack of information from fossil lineages has the potential to become much more problematic (*Blomberg, Rathnayake & Moreau, 2020*). Fortunately, if their phylogenetic placement is unambiguous, it would be straightforward to include trait measurements from fossils in the type of analysis presented here.
In that case, fossil species would be included as either terminal edges of zero length at the node, or along the branch, in or onto which they've been placed, *or* as extinct relatives of extant taxa. (This will depend on whether or not we've hypothesized that the fossil species is a direct ancestor of a living taxon or belongs on a separate branch.)

## CONCLUSIONS

Numerous phylogenetic comparative methods have been developed in recent years. Many of these are designed to model heterogeneity in tempo and mode across the nodes and branches of a phylogeny. Here, I present a simple penalized-likelihood method for modeling and graphing variation in the tempo of evolutionary change across the edges of a phylogeny. The method requires that we (arbitrarily) select a penalty or smoothing coefficient, $\lambda$, that will in turn greatly affect our inference about the magnitude of rate heterogeneity among branches in our tree. Nonetheless, I suggest that a cross-validation

approach (following *Smith & O'Meara (2012)*) might prove useful for more rigorous selection of this coefficient.

## ACKNOWLEDGEMENTS

Thanks to J. Berv, L. Harmon, and L. Schmitz for useful comments on this project as it was developed. S. Blomberg and an anonymous reviewer both provided incredibly insightful comments on an earlier version of this article. Special thanks to S. Blomberg for identifying the relationship between the method of this project and the SABR model from finance.

### Funding

This research was funded by grants from the National Science Foundation (DBI-1759940) and FONDECYT, Chile (1201869). The funders had no role in study design, data collection and analysis, decision to publish, or preparation of the manuscript.

### Grant Disclosures

The following grant information was disclosed by the authors:
National Science Foundation: DBI-1759940.
FONDECYT, Chile: 1201869.

### Competing Interests

The author declares that they have no competing interests.

### Author Contributions

- Liam J. Revell conceived and designed the experiments, performed the experiments, analyzed the data, prepared figures and/or tables, authored or reviewed drafts of the paper, and approved the final draft.

### Data Availability

Data and code to reproduce all the analyses of this study are available at GitHub: https://github.com/liamrevell/multirateBM.figures.

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
