# Peer review of "A variable-rate quantitative trait evolution model using penalized-likelihood"

_PeerJ, doi:10.7717/peerj.11997_

## Round 0.1 · original submission · Minor Revisions

Although both reviewers were enthusiastic about your study, they identified a number of smaller issues that need to be addressed. Please pay attention to the annotated version of the manuscript provided by Reviewer 2.

Reviewer 1 ·

Basic reporting

The paper is well-written and describes a new method using Penalized Likelihood to infer variation in rates of BM across the tree. The paper adequately describes the approach and provides some exploration of its behavior.

Experimental design

The paper is well-written and describes a new method using Penalized Likelihood to infer variation in rates of BM across the tree. The paper adequately describes the approach and provides some exploration of its behavior.

Validity of the findings

Results are interesting and will be helpful for the reader interested in applying the method.

Additional comments

Lines 141 to 143: I understand this example is here just to show the overall expectation under this model when rates get higher. Would be good to include a warning of the huge variation in rates here. From 0.186 to 368. A rate of 368 is so fast that the phenotype value becomes almost a random draw because the multivariate normal distribution is extremely wide. This figure might also give the false impression that this method will estimate unrealistic rates of trait evolution, which I think is not the intention of the author. So, it might be important to say that.

Figure 3: The same comment made above applies here. Is it possible to provide an example with a rate variation that is closer to what we observe in empirical studies? The figure shows that the method is capable of recovering the correct distribution of rates (with lambda = 1), but it would be better to have an example with rate variation inspired by empirical data. For example, having a range of BM rates equal to the range estimated under some data set.

Lines 156 to 162: Alternatively, the author could have shuffled the rates of the branches in Figure 2 to produce a case of uncorrelated rates with the exact same rate variation. The result should be very similar, however.

Figure 5: The results here are interesting. It is possible to see a different behavior of the model when the rates are uncorrelated. The penalty of lambda quickly result in practically a homogeneous trait evolution model (i.e., a single BM rate applied to the whole tree). This rate seems to be very close to the average rate of the (although to see the true mean rate we would need to weigh it by the length of the branches). Any idea of why this is happening?

Figure 7: Please include the explanation for the black and gray dots in the Figure caption.

Figure 8: Please include the explanation for the black and gray dots in the Figure caption.

Lines 262 to 269: Here the author reiterates the results shown in the previous section. However, there is no discussion about which properties of the model are likely the reason behind this pattern. It would be helpful if the author could extend this topic and discuss in more depth why the model behaves in this way and the possible reasons behind the difference in the results when rates evolve through BM and rates are uncorrelated.

Lines 270 to 275: How this result compares with other approaches that are able to estimate varying rates of BM across a tree without predictor regimes? For example, Eastman et al. (2011 Evolution 65:3578-3589) approach of RJMCMC to fit varying rates of a BM in a phylogeny, implemented in geiger::rjmcmc.bm is able to sample the posterior distribution of rate regimes across the tree. Other options would be BAMMtraits (by Dan Rabosky and colleagues) and bayOU (by Josef Uyeda – but with restrictions to model a BM process instead of an OU). It would be interesting to compare the output from a RJMCMC approach with the Penalized Likelihood method.
Although the author makes it clear that this method is intended to explore the data, and RJMCMC is a very distinct approach, users will ask this question when choosing which approach to use and it would be very helpful to have the perspective from the author here. If the objective is an exploration of the rates of trait evolution across the tree, then a data-driven approach such as RJMCMC can provide an alternative result like the one described here. A contrast and comparison with other approaches already available is necessary here.

About the ‘multirateBM’ program. Currently, it seems to only be available through the github repository for “phytools” and not via the standard installation using “install.packages”. This needs to be noted in the paper. It would be also very helpful to improve the help page for the function. More specifically, more details about the use of lambda (in the “Details” section). The help function also does not have any “Examples”, which are very helpful for the users. It seems that the same empirical analysis performed here can be added as an example in this function (the data seems to be also available in the package).

Finally, scripts for the plots and simulation done here have not been made available (or their availability was not made clear enough in the paper).

·

Basic reporting

Clear English, perhaps a little too conversational. Mostly sufficient references, but not in relation to some important concepts (such as cross-validation, previous studies of similar models in the financial literature.)

Experimental design

Definitely within the scope of the journal. The research question is theoretical: Can we build evolutionary models with varying rates of evolution over the tree? The paper certainly brings to light a knowledge gap, and is a step closer to filling that gap. Mostly rigorous and very technical. The functions are available in the author's R package, which is leading software for these kinds of analyses.

Validity of the findings

The findings are valid, as far as they go. These models are difficult to fit and suffer from "survivor bias:" Our conclusions are only based on extant taxa. The full evolutionary history of the study organisms might lead us to different conclusions. Unfortunately, a good fossil record is rarely available! This is a problem with the entire field and I do not expect this paper to correct that misconception.

Conclusions are justified, so far as theoretical studies can be. The statistical techniques are well-known and sound. The conclusions are well stated and link back to the original model.

Additional comments

Interesting paper! It could lead to even better work. Please see my attached document.

---

## Round 0.2 · accepted · Accept

The modifications based on the suggestions by the reviewers improved considerably the manuscript (which was already really interesting). I think this will be a really useful tool to researchers in the area.